# Effects of Copy Number Variations in the Plectin (*PLEC*) Gene on the Growth Traits and Meat Quality of Leizhou Black Goats

**DOI:** 10.3390/ani13233651

**Published:** 2023-11-25

**Authors:** Ke Wang, Yuelang Zhang, Xiaotao Han, Qun Wu, Hu Liu, Jiancheng Han, Hanlin Zhou

**Affiliations:** 1Zhanjiang Experimental Station, Chinese Academy of Tropical Agricultural Sciences, Zhanjiang 524013, China; lp_wangke@163.com (K.W.); xthan0521@163.com (X.H.); wuqun.2006@163.com (Q.W.);; 2Hainan Institute, Zhejiang University, Sanya 572024, China; zhangyuelang@zju.edu.cn

**Keywords:** *PLEC*, copy number variation, goats, growth traits, meat quality

## Abstract

**Simple Summary:**

As the global population continues to grow and the demand for unique meat flavor increases, goat meat has emerged as a sustainable choice for meat consumption. However, improving goat meat production and quality has become a pressing challenge for the industry. The *PLEC* gene has multiple regulatory effects on muscle development. Our previous study revealed that the *PLEC* gene in Leizhou black goats has numerous structural variations. In this study, three CNVs were identified in 417 Leizhou black goats. *PLEC* CNV-1 showed a significant association with chest circumference, body weight, carcass weight, the cross-sectional area of the longissimus dorsi lumbar muscle, and shear stress (*p* < 0.05). The gain type is significantly associated with *PLEC* expression in muscle (*p* <0.01). These findings suggest that *PLEC* CNV-1 plays a key role in the growth and muscle development of Leizhou black goats, highlighting its potential as a valuable molecular marker for genetic improvement in goat breeding programs.

**Abstract:**

The plectin (*PLEC*) gene is crucial in regulating muscle development and maintaining the cytoskeleton. An abnormal expression of *PLEC* can lead to muscle atrophy and muscular dystrophy. In a previous study, we found that Leizhou black goats exhibit abundant structural variations in the *PLEC* gene. However, the genetic effects of these variations on growth traits and meat quality in goats are not fully understood. In this study, three *PLEC* copy number variations (CNVs) were identified in a population of 417 Leizhou black goats, using quantitative polymerase chain reaction (qPCR) technology. Population distribution analysis revealed a high abundance of various types of these three CNVs. *PLEC* mRNA was found to be highly expressed in muscle tissue and remained consistently high from 1 month to 24 months after birth. Specifically, the gain type of CNV-1 (chr14: 81056401-81064800) showed a significant association with *PLEC* mRNA expression in muscle tissue (*p* < 0.01). The sequence of CNV-1 in *PLEC* shares similarities with three domain superfamilies associated with muscle development and skin disease. Furthermore, there were significant differences in chest circumference, body weight, carcass weight, the cross-sectional area of the longissimus dorsi lumbar muscle, and shear stress between different types of CNV-1 (*p* < 0.05). Notably, goats with the CNV-1 gain type demonstrated better phenotypic values compared to those with loss and normal types. These findings suggest that *PLEC* CNV-1 could play a crucial role in the growth and muscle development of Leizhou black goats, making it a potential marker for assisted selection in goat breeding.

## 1. Introduction

Compared to other domestic species, goats are considered the ideal animal model for studying climate change due to their high thermal and drought resilience, ability to survive on limited pastures, and high disease resistance [1]. They also require less feed and have a smaller carbon footprint, making goat meat (chevon) a more sustainable choice for meat consumption [2]. It is lower in fat and cholesterol compared to other red meats, such as beef or lamb, and it offers a tasty and nutritious alternative to other red meats.

The plectin (*PLEC*) gene is a protein-coding gene that performs a variety of functions. Dysregulation or mutations in the *PLEC* gene can have significant consequences, leading to a deficiency or malfunction of the plectin protein. Plectin is responsible for linking the cytoskeleton (the internal cellular framework) [3] to the extracellular matrix (the external support structure) [4]. This linkage is essential for maintaining the stability and strength of muscle cells. On one hand, *PLEC* regulates canonical Wnt signaling-mediated skeletal development by stabilizing Dvl-2 and downregulating the cellular autophagic degradation system [5]. The expression pattern of *PLEC* in the longissimus dorsi muscle at three key developmental stages is significantly different [6]. On the other hand, mutations in the *PLEC* gene result in weakened muscle fibers and progressive muscle degeneration, which leads to muscle weakness [7], difficulty in movement [8], and the eventual loss of muscle function [9]. Additionally, mutations in the *PLEC* gene can also contribute to other muscle-related conditions, such as limb–girdle muscular dystrophy [10] and myofibrillar myopathy [11].

CNV (Copy Number Variation) refers to a type of genetic variation in which the number of copies of a specific DNA segment varies among individuals in a population [12]. It involves the gain or loss of DNA segments that are larger than 1 kilobase in size. CNVs can occur in both coding and non-coding regions of the genome and can have significant impacts on gene expression and phenotypic variation [13,14]. CNVs can arise through various mechanisms, including DNA replication errors, unequal crossing over during meiosis, and genomic rearrangements. They can be inherited from parents or occur de novo in an individual. CNVs can affect gene dosage, disrupting the normal balance of gene expression [15]. When a CNV results in the loss of one copy of a gene, it can lead to haploinsufficiency, where the remaining copy of the gene is insufficient to maintain normal function. Conversely, CNVs that result in the gain of gene copies can lead to gene overexpression, potentially altering cellular processes and contributing to disease [16,17]. CNV is widely used in livestock. On the one hand, the analysis of CNV differences between different individuals can identify CNV regions associated with traits and use them as selective markers for genome selection and for genetic improvement [18,19,20]. On the other hand, comparing the differences in CNV between infected and uninfected individuals allows for the identification of regions of CNV associated with disease resistance [21].

The Leizhou black goat (LZBG), the primary meat goat breed in the tropical region of China, frequently displays congenital muscle atrophy and malnourished phenotypes in its offspring, including slow growth and difficulty standing. These symptoms result in reduced productivity, decreased feed efficiency, weakened musculoskeletal systems, impaired reproductive performance, and decreased market value [22,23,24]. In the unpublished annotation results of our whole-genome resequencing of the LZBG breed, we discovered that the LZBG breed has numerous structural variations (SVs) in the *PLEC* gene, which plays a vital role in regulating muscle development. Therefore, the objective of this study was to investigate the potential effects of copy number variations (CNVs) in the *PLEC* gene of goats on the growth traits and meat quality of the LZBG breed. These findings will contribute to a better understanding of the impact of the PLEC gene on goat muscle and provide a basis for genetic improvement of goats.

## 2. Materials and Methods

### 2.1. Samples and Data Collection

Under the same feeding and management conditions [25,26], ear tissues were randomly collected from 417 female goats (720 ± 1.47 days old; means ± SE) at the Leizhou Black Goat Breeding Farm. The growth traits of these goats, including body height (BH), body oblique length (BOL), chest circumference (CC), body weight (BW), and cannon circumference (CAC), were measured and recorded. From the 417 goats, 80 individuals were randomly selected for slaughter, and various meat quality variables were measured and recorded, including carcass weight (CW), a cross-sectional area of the longissimus dorsi lumbar muscle (CALM), water loss rate (WLR), water holding capacity (WHC), and shear stress (SS). Additionally, the heart, liver, kidney, small intestine, large intestine, back skin, subcutaneous fat, abdominal fat, longissimus dorsi muscle, and gluteofemoral biceps of 12 adult female goats were randomly collected for gene expression analysis. Furthermore, 12 samples of longissimus dorsi muscle were collected from LZBGs at 1 month, 6 months, 12 months, and 24 months of age for expression profiling. All samples were immediately stored in RNAlater^TM^ (Beyotime, Shanghai, China) after collection and then transferred to storage at −80 °C. All experimental procedures were approved by the Review Committee for the Use of Animal Subjects of the Chinese Academy of Tropical Agricultural Sciences, and the animal experimentation was performed in accordance with the guidelines of the ethics commission (CATAS-20230017ZES).

### 2.2. Bioinformatics Analysis

The amino acid sequences of the PLEC protein from various species were obtained from the NCBI database (https://www.ncbi.nlm.nih.gov/protein; accessed on 14 July 2023). We obtained the predicted proteins for the goats bos indicus and bos mutus. In order to display the similarities and differences of amino acid sequences more comprehensively, we selected the isoform with the highest length and gene coverage among different species for comparison. The species include Capra hircus (XP_017914185.1), Ovis aries (XP_027829099.2), Homo sapiens (NP_000436.2), Mus musculus (NP_001157012.1), Bos taurus (XP_024857673.1), Rattus rattus (XP_032773070.1), Sus scrofa (XP_020946022.1), Bos indicus (XP_019828947.1), Bos mutus (XP_014334314.1), and Bubalus bubalis (XP_006054396.4). The MUSCLE program in MEGA-11 (V11.0.13) the Neighbor-Joining (NJ) method were used to compare the multiple sequences and construct the phylogenetic tree. To comprehensively investigate the structural features and functional motifs of the PLEC protein, we employed the MEME suite (https://meme-suite.org/; accessed on 25 July 2023) [27]. Utilizing this tool, we conducted an analysis of the PLEC protein motifs across diverse species. This approach allowed us to gain insights into the conserved elements and variations in the protein’s motif architecture, contributing to a more comprehensive understanding of its biological significance. Additionally, the conservative domain structure and function of the PLEC protein were analyzed through the CDD NCBI (https://www.ncbi.nlm.nih.gov/cdd/ (accessed on 25 July 2023)) [28].

### 2.3. Genomic DNA and Total RNA Extraction

The genomic DNA was extracted from ear tissue using the Animal Tissues/Cells Genomic DNA Extraction Kit (Solarbio, Beijing, China). The concentration of the DNA was determined using a Nanodrop One spectrophotometer (Thermo Fisher, Waltham, MA, USA). The DNA was diluted to a standard concentration of 20 ng/µL and stored at −20 °C. Total RNA was extracted using the Trizol method (Solarbio, Beijing, China), and the first-strand cDNA was synthesized by PrimeScript™ RT Reagent Kit (Takara, Tokyo, Japan) following the manufacture’s protocol. 

### 2.4. Primer Design

According to the Vargoats Database (https://www.goatgenome.org/; accessed on 28 July 2023) [29], Goat Pan-genome Database (http://animal.omics.pro/; accessed on 28 July 2023) [30], and the unpublished LZBG whole genome sequencing (WGS) results, we identified three CNVs on the goat *PLEC* gene, named CNV-1 (ARS1_chr14: 81056401-81064800), CNV-2 (ARS1_chr14: 81078401-81082100), and CNV-3 (ARS1_chr14: 81098001-81100400). Subsequently, six primer pairs were designed for amplification using the Primer-BLAST tool (https://www.ncbi.nlm.nih.gov/tools/primer-blast/; accessed on 1 August 2023). *MC1R* and *GAPDH* were used as reference genes [31]. The relevant primers are listed in Appendix A.

### 2.5. Copy Number Analysis, mRNA Expression, and Statistical Analysis

Genomic DNA extracted from ear tissues was used for qPCR copy number analysis, following the protocol described by Weaver et al. [32]. The total qPCR reaction system consisted of 10 μL, comprising 5 μL of 2 × SYBR Premix Ex Taq (Takara, Japan), 0.5 μL of DNA, 0.5 μL of each primer, and 3.5 μL of ddH_2_O. The reaction conditions and thermal profiles were as previously described by Chen et al. [31]. Relative copy number (RCN) was calculated from the ΔΔ*Cq* value using the following formula [32]: Relative copy number (RCN) = 2^−ΔΔ*Cq*^

Copies per diploid genome = 2 × 2^−ΔΔ*Cq*^


The CNVs of the *PLEC* gene were classified into three types: loss (=1), normal (=2), and gain (>3).

The cDNA from different tissues was used to analyze *PLEC* mRNA expression. PCR amplification was performed following the protocol described by Wang et al. [33]. The internal control gene, *GAPDH*, and relative expression of each gene were determined using the 2^−ΔΔCt^. To assess the significance of the relationship between Copy Number Variation (CNV) genotypes and traits, we conducted statistical analyses using SPSS software (V25.0, Chicago, IL, USA). We employed a one-way analysis of variance (ANOVA) for this purpose. Additionally, we applied the same statistical approach, specifically one-way ANOVA, to determine the effect of CNV genotypes on mRNA expression levels. The analysis followed a linear model, as previously described by Wang et al. [34]: Y*_ijk_* = α*_i_*+ β*_j_* + e*_ijk_* + *u*

Here, Y*_ijk_* represents the evaluation of traits at the *i*th level of the fixed factor age (α*_i_*) and *j*th level of the fixed factor genotype (β*_j_*); *u* represents the overall mean; and e*_ijk_* is the random error. 

## 3. Results

### 3.1. Biological Evolution and Conservation Analysis of the PLEC

The amino acid sequences of the PLEC protein were compared and analyzed in 10 animal species. The structures of the PLEC protein was observed to be highly conserved in nine species, with the exception of yaks (Figure 1A). In 10 species, the PLEC protein exhibited three important motifs (Figure 1B). Phylogenetic tree analysis revealed a close relationship between goats, sheep, cattle, and other ruminants (Figure 1C). Since only CNV-1 of the three CNVs is located in the exon region, the *PLEC* CNV-1 region protein structure was analyzed using NCBI CDD. The results demonstrated that the CNV-1 region contained the calponin homology (CH) domain superfamily (cl00030), spectrin repeats superfamily (cl02488), and plectin repeat superfamily (pfam00681) (Figure 2). The goat *PLEC* gene encodes a total of 4556 amino acids, and the CNV-1 region contains a total of 2739 amino acids, including exon 19, which is the longest exon, and part of the 18th exon, accounting for 60.12% of the total peptide chain length.

### 3.2. Distribution of CNVs of PLEC in the LZBGs

The difference in the copy numbers of *PLEC* can be identified in the qPCR loop by the threshold number of times, which is derived from the result of the reference gene. The distribution of the *PLEC* copy number variation in the LZBG population is shown in Figure 3. The copy number range mainly ranges from 1 to 8 copies in CNV-1, 1 to 6 copies in CNV-2, and 1 to 12 copies in CNV-3. The highest proportion (63.86%) of the copy numbers in CNV-3 is the normal type (copies = 2), while the copy numbers of the loss type (copy < 2) occur less frequently in all three CNVs. The frequency of the gain type (copies ≥ 3) in CNV-2 is even higher than that of the normal type, reaching 48.56%.

### 3.3. Analysis for the PLEC Expression and Its Associations with Different CNVs in the LZBGs

Via analysis, *PLEC* expression was detected in 10 tissues of the LZBGs. As shown in Figure 4A, *PLEC* mRNA was widely expressed in adult goat tissues. Among these 10 tissues, the expression levels of *PLEC* mRNA in the heart and longissimus dorsi muscle were significantly higher than in other tissues (*p* < 0.05). PLEC expression was also much higher in the gluteofemoral triceps and back skin compared to fat and the gut (*p* < 0.05). Figure 4B shows the temporal expression detection of the longissimus dorsi muscle, indicating that *PLEC* expression remained consistently high from 1 month to 24 months after birth, without significant changes. These results suggest that the *PLEC* gene may play a crucial role in the development of muscles and skin. Additionally, we examined the relationship between *PLEC* CNVs and the *PLEC* mRNA expression levels in the LZBGs. In both muscle tissues, the mRNA expression of the *PLEC* gene was significantly associated with CNV-1 gaintypes (Figure 4C, *p* < 0.01). However, there was no significant association between *PLEC* expression and copy numbers for CNV-2 and CNV-3 (Figure 4D,E).

### 3.4. Association Analysis between the PLEC CNVs and Traits

The association analysis between CNVs in the *PLEC* gene and growth traits revealed that CNV-1 types were significantly associated with chest circumference and body weight (*P* < 0.05). Goats with the gain type of CNV-1 exhibited better phenotypic values compared to those with loss and normal types (Table 1). CNV-2 (Appendix A) and CNV-3 (Appendix A) did not have a significant effect on growth traits in LZBG goats. 

Regarding carcass traits and meat quality, different types of the CNV-1 showed significant differences in measurements of carcass weight, the cross-section area of longissimus dorsi lumbar muscle, and shear stress. Goats with the gain type of the CNV-1 showed better phenotypic values compared to those with loss and normal types (Table 1). Different types of the CNV-2 were also significantly associated with the means of carcass weight and shear stress (Appendix A), while no significant difference was found between CNV-3 types and any trait mean (Appendix A).

## 4. Discussion

Goat meat is a rich source of high-quality protein, essential vitamins, and minerals. From an environmental perspective, goats can efficiently convert grass and other forage into nutrient-rich meat, minimizing waste, and utilizing resources effectively. In order to introduce new genetic traits and improve the overall productivity and adaptability of Chinese goats, goats with desirable meat traits are imported from other regions or countries and crossed with local Chinese goats. This has become the prevailing breeding practice for goat meat in China. But many Chinese goat breeds have become genetically homogeneous due to limited breeding practices and a focus on specific traits, such as high milk yield or high meat quality. Leizhou black goat meat is famous for its tenderness, juiciness, and rich flavor. At the same time, as a less-studied Chinese local goat breed, LZBGs possesses rich genetic diversity and frequently displays congenital muscle atrophy and malnourished phenotypes in its offspring, including slow growth and difficulty standing. Abnormalities in the *PLEC* gene are known to be one of the key factors causing congenital muscular dystrophy and malnutrition [7,8], and we have identified a significant number of genetic variations in the *PLEC* gene [35]. Therefore, we conducted a series of analyses on three copy number variations (CNVs) on the *PLEC* gene to explore their potential impact on economic traits regarding the production of Leizhou goats in this study.

Firstly, we analyzed the protein structure and function of the *PLEC* gene. The PLEC protein motif and structure are highly conserved in sheep, cattle, and goats, suggesting stable inheritance and potential roles in animal growth and development. In addition, the plectin protein plays a crucial role in maintaining cytoskeletal structure. The non-conserved protein structure and motif of plectin in yaks may be attributed to the provisional nature of yak sequences. Typically, the stronger the function of a gene, the greater its conservation across species [36]. Specifically, CNV-1 overlaps with a segment of the PLEC exons. This structural variant, CNV, encompasses certain exons, potentially influencing various aspects such as the protein coding, transcription regulation, splicing regulation, RNA stability, and transport functions of the gene [12,37]. The specific impact is contingent upon the sequence of the exon and its position within the gene regulatory network. The sequence of *PLEC* CNV-1 overlaps with the calponin homology domain superfamily, spectrin repeat superfamily, and plectin repeat superfamily. Previous studies have shown that C alponin H omology domains are actin filament (F-actin) binding motifs, which can exist as a single copy or in tandem repeats, increasing binding affinity [38,39]. The spectrin repeat superfamily and plectin repeat superfamily are present in several proteins that are associated with cytoskeletal structure and skin pemphigus [9,40]. All of these proteins are closely related to the known function of the *PLEC*. CNVs may potentially affect gene expression by altering gene dose and transcriptional structure [41]. Alterations in the copy number of this extensive coding sequence suggest that, irrespective of the copy number itself, the presence of multiple copies may lead to the deterioration of the gene. To gain insights into the potential impact, the SWISS-MODEL was employed to predict the protein structure based on the amino acid sequence encoded by CNV-1 (https://swissmodel.expasy.org/; accessed on 16 November 2023) [42]. Notably, our analysis revealed that the amino acid sequence derived from the CNV-1 region (Appendix A) might give rise to a distinct protein structure compared to the complete goat plectin protein (Appendix A). However, current research methods do not provide sufficient means to conduct experiments on goat plectin protein directly. Therefore, we postulate that the primary function of CNV-1 could involve the regulation of PLEC gene expression by influencing its transcriptional structure. This hypothesis necessitates further investigation into the underlying mechanisms of action.

Secondly, we detected the population distribution of three CNVs. It was found that these three CNVs are highly abundant. Next, we analyzed the expression of *PLEC* in LZBGs and its association with different CNVs. Consistent with previous studies, *PLEC* exhibited high expression in the skin and muscle. Furthermore, it maintained a high expression level in the muscles of goats after birth, without significant changes with age. This stable high expression is required for the role of the PLEC protein in the organization and maintenance of the cytoskeleton. The PLEC protein acts as a bridge between different structures inside the cell, helping the cell maintain shape and structural stability [43]. The relationship between CNVs and *PLEC* mRNA expression revealed that gain types of CNV-1 significantly promote *PLEC* gene expression. This impact was only observed in CNV-1, while no impact on the expression of the *PLEC* gene was detected in CNV-2 and CNV-3, possibly due to their location within intronic regions. 

Lastly, we examined the influence of these CNVs on economic traits in goats. For CNV-1, goats with loss types exhibited lighter body weight and smaller chest circumferences compared to those with gain and normal types. Carcass weight, the cross-sectional area of the longissimus dorsi lumbar muscle, and shear stress were significantly different among the three types. On the other hand, the different types of CNV-2 and CNV-3 only differ significantly in carcass weight and shear stress. Therefore, we believe that CNVs in the *PLEC* gene play an important role in regulating the growth traits and meat quality of LZBGs. This effect is primarily attributed to the multiple regulatory effects of *PLEC* on muscle development, as other researchers have demonstrated in previous studies [5,44,45,46,47].

Yin et al. [5] discovered that *PLEC* plays a crucial role in promoting the differentiation and proliferation of C2C12 myoblasts, while inhibiting their apoptosis. On one hand, PLEC can interact with Dishevelled-2 (Dvl-2) and form a protein complex, which activates the canonical Wnt signaling pathway. The Wnt signaling pathway regulates the activation and proliferation of muscle progenitor cells and promotes the differentiation of myoblasts into mature muscle fibers by activating specific transcription factors such as MyoD and Myogenin [44]. A dysregulation of the Wnt signaling pathway can lead to muscle disorders and impairments in muscle function [45]. On the other hand, PLEC prevents ubiquitination by stabilizing Dvl-2, thereby reducing the level of LC3-labeled Dvl-2 and antagonizing the autophagy system. The inhibition of autophagy results in incomplete skeletal muscle recovery and disrupts muscle regeneration [46,47]. In this study, the traits of body weight, chest circumference, carcass weight, and the cross-sectional area of the longissimus dorsi muscle were directly influenced by muscle mass, leading to significant differences in the means for these traits between the different CNV types. Additionally, *PLEC* also regulates the expression of atrophy-related genes (atrogin-1 and MuRF-1) to rescue muscle atrophy [48]. It regulates the mechanical properties of myoblasts such as cellular stiffness, the ability of adhesion, and contractile force [49]. This may explain the differences in muscle shear stress observed with different CNV types.

Taking these findings together, we speculated that *PLEC* CNVs influence the growth traits and meat quality of LZBGs by influencing *PLEC* expression levels in muscles. In future research, we will further investigate the molecular regulatory mechanism of *PLEC* genetic variations on meat quality traits. 

## 5. Conclusions

In this study, we conducted a comprehensive investigation into three CNVs within the *PLEC* gene. Our analysis focused on their distributions, impact on *PLEC* expression, and their significance in relation to economic traits in the LZBG population. The findings revealed that distinct types of CNV-1 can indeed influence *PLEC* gene expression, exerting a significant impact on both meat quality and growth traits. This discovery holds promising implications for the development of molecular markers and establishes a crucial foundation for genetic advancements in goat breeding, facilitating targeted improvements in key economic traits.

## Figures and Tables

**Figure 1 animals-13-03651-f001:**
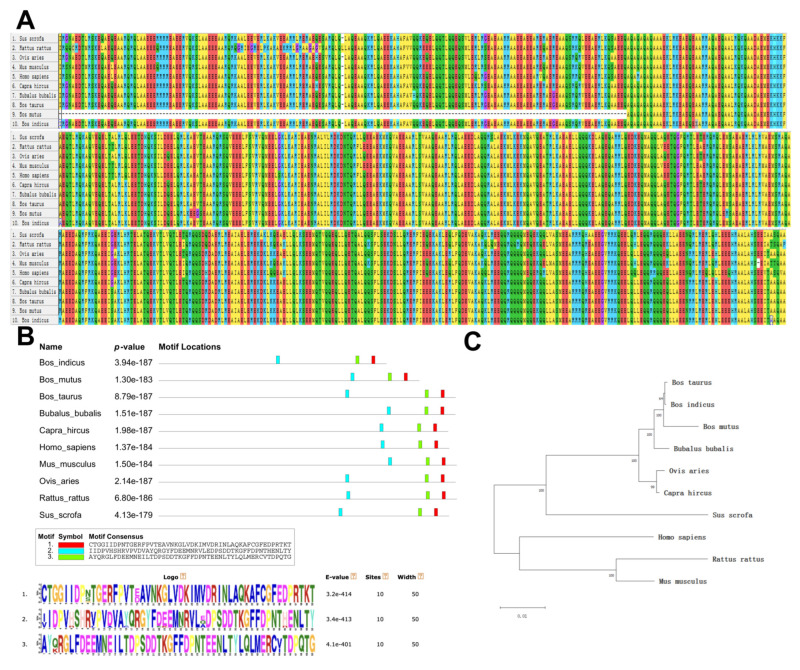
Biological evolution and the conserved domains of the *PLEC* gene. (**A**) Multiple sequence alignment of *PLEC* for 10 species. (**B**) Motif structural analysis for PLEC among 10 species. (**C**) Phylogenetic tree analysis for the *PLEC* gene among 10 species.

**Figure 2 animals-13-03651-f002:**
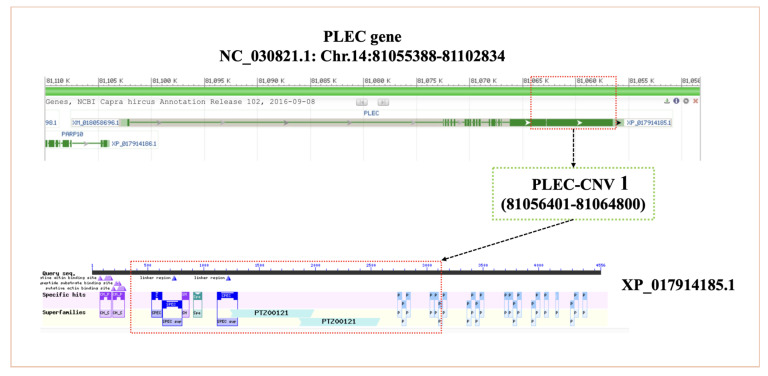
The location of the PLEC-CNV1 and the schematic diagram overlaps between CNV-1-related protein region and conserved domains in the *PLEC* protein sequence.

**Figure 3 animals-13-03651-f003:**
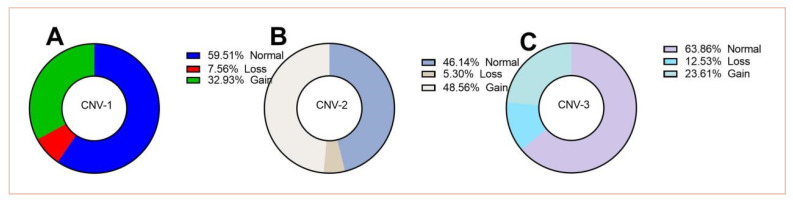
Distribution of CNV-1 (**A**), CNV-2 (**B**), and CNV-3 (**C**) of the PLEC gene in the LZBGs. Loss, Normal and Gain were defined as copy number <2, =2, or ≥3, respectively.

**Figure 4 animals-13-03651-f004:**
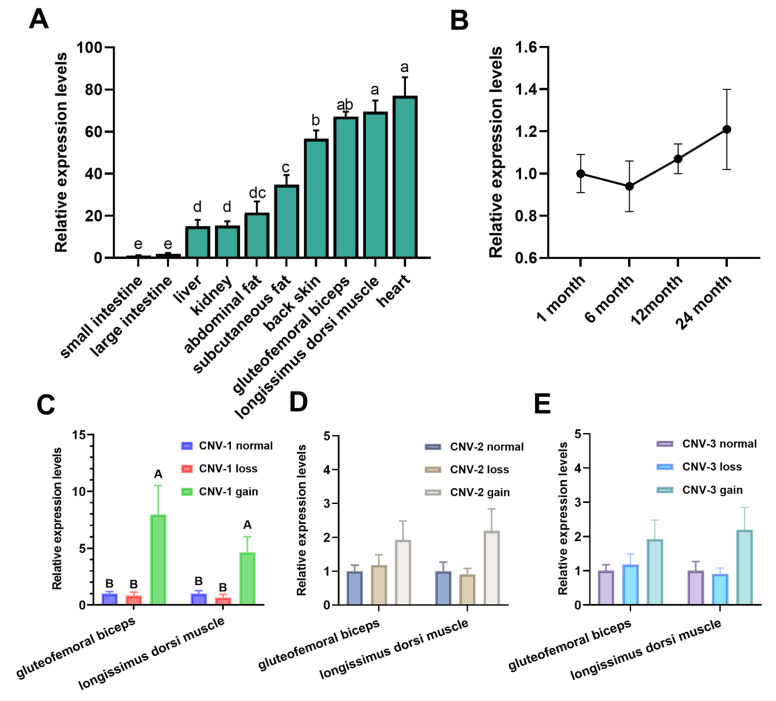
Comparison analysis of the *PLEC* expression levels. (**A**) The *PLEC* mRNA expression profile in 10 tissues of the 12 adult female LZBGs. (**B**) Comparison of the *PLEC* mRNA levels among different times in the longissimus dorsi muscle of 12 LZBGs. (**C**–**E**), Comparison of the PLEC expression levels among different CNVs and different genotypes in longissimus dorsi muscle and gluteofemoral triceps in the LZBGs. Different letters (a–e) represent significant differences (a–e: *p* < 0.05; A, B: *p* < 0.01).

**Table 1 animals-13-03651-t001:** Association analysis between the traits and CNV-1 in the goat *PLEC* gene. Different letters (a–c) represent significant differences (*p* < 0.05).

Growth Traits	CNV Types (Mean ± SE)			*p* Values
	Loss (1 Copy) (*n* = 31)	Normal (2 Copies) (*n* = 244)	Gain (≥3 Copies) (*n* = 135)	
body height (BH, cm)	50.90 ± 0.19	51.64 ± 0.10	51.43 ± 0.18	0.165
body oblique length (BOL, cm)	53.94 ± 0.30	55.32 ± 0.12	54.92 ± 0.09	0.548
chest circumference (CC, cm)	56.37 ^b^ ± 0.44	57.48 ^b^ ± 0.13	61.09 ^a^ ± 0.15	0.016
body weight (BW, kg)	18.09 ^b^ ± 0.36	19.27 ^b^ ± 0.06	21.63 ^a^ ± 0.10	0.039
cannon circumference (CAC, cm)	7.31 ± 0.02	7.20 ± 0.01	7.18 ± 0.03	0.644
**Meat Quality**	**Loss (1 Copy)** **(** * **n** * ** = 12)**	**Normal (2 Copies)** **(** * **n** * ** = 41)**	**Gain (≥3 Copies)** **(** * **n** * ** = 27)**	
carcass weight (CW, kg)	8.64 ^c^ ± 0.24	9.77 ^b^ ± 0.16	11.62 ^a^ ± 0.01	0.011
cross-sectional area of *longissimus dorsi lumbar* muscle(CALM, cm^2^)	7.04 ^b^ ± 0.15	7.95 ^b^ ± 0.15	8.76 ^a^ ± 0.16	0.038
water loss rate (WLR, %)	4.88 ± 0.13	4.83 ± 0.04	4.91 ± 0.12	0.492
water holding capacity (WHC, %)	4.77 ± 0.07	4.81 ± 0.03	4.90 ± 0.03	0.367
shear stress (SS, N)	54.63 ^a^ ± 0.57	47.98 ^b^ ± 0.17	45.17 ^c^ ± 0.33	0.003

## Data Availability

The data that support the findings of this study are available on request from the corresponding author, J.H. upon reasonable request.

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
