# Peer review of "Effects of Copy Number Variations in the Plectin (PLEC) Gene on the Growth Traits and Meat Quality of Leizhou Black Goats"

_animals, 2023, doi:10.3390/ani13233651_

Round 1

Reviewer 1 Report

Comments and Suggestions for Authors

This is an interesting paper with the potential to be published. I would like however to see some issues addressed.

In all the paper check referencing style

Line 69 In our previous study (give the reference)

Line 90-91 . All samples 90 were immediately stored in RNA later after collection and then transferred to storage at - 91 80 °C (please Review this sentence)

Figure 1 difficult to read. I would suggest breaking it down in several figures to increase size and also quality needs to be improved.

Line 141-142 The linear model was established as previously described by Akhatayeva et al. (2022). I have checked the linear model used by the reference gives and the variables are completely different. I, therefore, ask you to describe the model used in your analyses identifying factors and variables and also stating how you checked the prerequisites for this model.

Line 145 11 species?? I have counted 10 - goat, sheep, man, rat, mouse, buffalo, yak, Taurus cattle, zebu cattle, and pig. Therefore in line 146, 9 and 10 may also need to change,…. Check

Figure 3 Legend should be different letters (a,b,c,d,e) in the bars associated with the muscles….

Figure 3 statistical tests comparing C, D and E this is probably a two way ANOVA you need to refer to it in your method. Also, what is the overall test P-value. And the p-values for factor analysis? Did you check interactions? Did you check the prerequisites ? normal distribution of residuals, homogeneity of variances?

Table 1. These are also ANOVAs. You need to be specific also in methodology in relatition to the statistical approach. Need to state that different letter in superscript……etc…. also don’t use capital letters in superscript,… be consistent. Also rearrange text sin such a way to avoid tables to be broken in 2 pages.

Line 226-227 you probably want to say that your results tally those of the author in the reference.

Line 231 careful with stats terminology. You did not check for correlations but instead, you have tested significant differences between means. Check throughout the paper.

Author Response

Dear Reviewer,

Thank you for taking the time to review our manuscript and for providing valuable feedback. We appreciate your thorough evaluation of our work and your suggestions for improvement. We have carefully considered your comments and have made the necessary revisions accordingly. We uploaded the revised version, all the changes are marked in red.

[Reviewer's Comment 1]

In all the paper check referencing style.

[Author's Response/Revision]

Thanks to the editor for formatting changes to the references in our manuscript, we have updated some references and referencing style in the revision.

[Reviewer's Comment 2]

Line 69 in our previous study (give the reference)

[Author's Response/Revision]

Sorry for the lack of clarity here, in fact, our whole gene resequencing project for LZBG is not fully completed and is not yet published. We will complete and release the whole genome resequencing results of LZBG in the next 3-6 months, and publish the corresponding article. Therefore, we have added a supplementary description in the revision.

[Reviewer's Comment 3]

Line 90-91. All samples were immediately stored in RNA later after collection and then transferred to storage at -80 °C (please Review this sentence)

[Author's Response/Revision]

The use of RNAlater as a protective agent to inhibit RNase activity may cause ambiguity here, and we have added its information in the revised revison.

[Reviewer's Comment 4]

Figure 1 difficult to read. I would suggest breaking it down in several figures to increase size and also quality needs to be improved.

[Author's Response/Revision]

According to your suggestion, we re-split Figure 1 and raised the dpi of the picture.

[Reviewer's Comment 5]

Line 141-142 The linear model was established as previously described by Akhatayeva et al. (2022). I have checked the linear model used by the reference gives and the variables are completely different. I, therefore, ask you to describe the model used in your analyses identifying factors and variables and also stating how you checked the prerequisites for this model.

[Author's Response/Revision]

About the linear model, we consulted Professor LAN Xianyong during the analysis process, and he assigned Dr. Akhatayeva to guide us in building the analysis model. In fact, the model we used was based on the model in another article about CNV that Dr. Akhatayeva participated in. However, in the process of writing, we mistakenly quoted another article. In order to correct this mistake, we carried out a detailed description of the model and corrected the reference in the revision.

[Reviewer's Comment 6]

Line 145 11 species?? I have counted 10 - goat, sheep, man, rat, mouse, buffalo, yak, Taurus cattle, zebu cattle, and pig. Therefore in line 146, 9 and 10 may also need to change,…. Check

[Author's Response/Revision]

We are very sorry for this mistake, our analysis was conducted in 10 species, and the gene is highly conserved in 9 species outside the yak. It has been corrected in the revision.

[Reviewer's Comment 7]

Figure 3 Legend should be different letters (a,b,c,d,e) in the bars associated with the muscles….

[Author's Response/Revision]

Based on your comments, we have made changes to the legend in Figure 3.

[Reviewer's Comment 8]

Figure 3 statistical tests comparing C, D and E this is probably a two way ANOVA you need to refer to it in your method. Also, what is the overall test P-value. And the p-values for factor analysis? Did you check interactions? Did you check the prerequisites? normal distribution of residuals, homogeneity of variances?

[Author's Response/Revision]

Thank you very much for your advice. In fact, we believe that the comparison between CNV genotypes and expression level (there is no comparison between different genotypes of different CNVS in the analysis) can be done by one-way ANOVA. And we had referred it in our revision. Normality and Lognormality Tests were carried out in the analysis process. In order to more intuitively represent this significant difference, we chose to use a bar chart to show the results. If you need confirmation, I can provide you with the data table in the analysis process in the future. (I couldn't add all 12 tables in total to the online box in the system).

[Reviewer's Comment 9]

Table 1. These are also ANOVAs. You need to be specific also in methodology in relatition to the statistical approach. Need to state that different letter in superscript……etc…. also don’t use capital letters in superscript,… be consistent. Also rearrange text sin such a way to avoid tables to be broken in 2 pages.

[Author's Response/Revision]

Thanks for your comments. We had modified it according to your suggestion in our revision.

[Reviewer's Comment 10]

Line 226-227 you probably want to say that your results tally those of the author in the reference.

[Author's Response/Revision]

This sentence has been corrected in the version to ensure accuracy.

[Reviewer's Comment 11]

Line 231 careful with stats terminology. You did not check for correlations but instead, you have tested significant differences between means. Check throughout the paper.

[Author's Response/Revision]

Thank you very much for your professional advice on statistics. We have made modifications to the use of this statistical terminology in the revision.

We would like to express our gratitude for your insightful comments, which have undoubtedly enhanced the quality and clarity of our manuscript. We believe that these revisions have significantly strengthened our work and addressed the concerns raised during the review process. Once again, we sincerely appreciate your time and effort in reviewing our manuscript. Your expertise and constructive feedback have been invaluable in shaping our research. We look forward to hearing from you regarding the final decision on our submission.

Reviewer 2 Report

Comments and Suggestions for Authors

The study presented by Wang et al. investigated the Effects of copy number variations of the PLEC gene on growth traits and meat quality of Leizhou black goat. The study keeps mentioning something like Our previous study revealed that the PLEC gene in Leizhou black goats has numerous structural variations", but I don't find which of the previous study reports this. Additionally, I even don't know where are those three CNV identified in the goat genome.

Author Response

Dear Reviewer,

Thank you for taking the time to review our manuscript and for providing valuable feedback. We appreciate your thorough evaluation of our work and your suggestions for improvement. We have carefully considered your comments and have made the necessary revisions accordingly. We uploaded the revised version, all the changes are marked in red.

[Reviewer's Comment 1]

The study keeps mentioning something like Our previous study revealed that the PLEC gene in Leizhou black goats has numerous structural variations", but I don't find which of the previous study reports this. Additionally, I even don't know where are those three CNV identified in the goat genome

[Author's Response/Revision]

We apologize for misleading you with our incomplete description. In fact, our whole gene resequencing project for LZBG is not fully completed and is not yet published. We will complete and release the whole genome resequencing results of LZBG in the next 3-6 months, and publish the corresponding article. Therefore, we have added a supplementary description in the revision. Meanwhile, in section 2.4. Primer Design, we have explained the specific CNV positioning information like “we confirmed the presence of three CNVs on the goat PLEC gene, named CNV-1 (chr14: 81056401-81064800), CNV-2 (chr14: 81078401-81082100) and CNV-3 (chr14: 81098001-81100400).”

We would like to express our gratitude for your insightful comments, which have undoubtedly enhanced the quality and clarity of our manuscript. We believe that these revisions have significantly strengthened our work and addressed the concerns raised during the review process. Once again, we sincerely appreciate your time and effort in reviewing our manuscript. Your expertise and constructive feedback have been invaluable in shaping our research. We look forward to hearing from you regarding the final decision on our submission.

Reviewer 3 Report

Comments and Suggestions for Authors

Dear Authors, I read your article carefully. My suggestion is that the article can be published after careful review.

In addition to the points reported directly in the attached PDF, I invite you to discuss in depth the impact of CNV1 on the molecular structure of the gene involved, as this CNV seems to involve a good part of the coding sequence.

Author Response

Dear Reviewer,

Thank you for taking the time to review our manuscript and for providing valuable feedback. We appreciate your thorough evaluation of our work and your suggestions for improvement. We have carefully considered your comments and have made the necessary revisions accordingly. We uploaded the revised version, all the changes are marked in red.

[Reviewer's Comment 1]

Line 69 you must add a reference to this quote.

[Author's Response/Revision]

Sorry for the lack of clarity here, in fact, our whole gene resequencing project for LZBG is not fully completed and is not yet published. We will complete and release the whole genome resequencing results of LZBG in the next 3-6 months, and publish the corresponding article. Therefore, we have added a supplementary description in the revision.

[Reviewer's Comment 2]

Line 79 State what this means: mean and standard deviation?

[Author's Response/Revision]

Corrections have been made as you suggested.

[Reviewer's Comment 3]

Line 97-102 I noticed that there are different isoforms of this gene: how do they differ? I suggest indicating that for the following species these are predicted proteins: goat, bos indicus and bos mutus. Furthermore, in some cases it is isoform 1, in others it is isoform 2: I don't think it is correct to compare two different isoforms with each other.

[Author's Response/Revision]

Thank you for your professional comments. We have added descriptive sentences to meet your requirements. In order to display the similarities and differences of amino acid sequences more comprehensively, we did not select the isoform 1 of PLEC in each species according to the order of expression abundance, but selected the isoform with the highest length and gene coverage among different species for comparison. If you insist that this comparison is not rigorous, we will then only compare isoform 1 across species.

[Reviewer's Comment 4]

Line 115-116 In this case it is necessary to indicate the name of the commercial kit

[Author's Response/Revision]

Corrections have been made as you suggested.

[Reviewer's Comment 5]

Line 118 if it is not published you must demonstrate the existence of these CNVs

Line 119 It is not clear whether these are results already observed, as described in line 69, or new results. Also do you discover or confirm?

Line 200 it is necessary to report which genomic assembly these positions refer to.

[Author's Response/Revision]

Thanks for your comments. We changed the words and phrases in the revision to be more accurate.

[Reviewer's Comment 6]

Line 141 As you rightly say, you used SPSS SOFTWARE, but it is important to report which statistical test you used.

[Author's Response/Revision]

Thanks for your comments. We have made changes as suggested by you and another reviewer.

[Reviewer's Comment 7]

Line 147 I believe that this observation is not due to the fact of a real evolutionary divergence but to the fact that the yak sequence is still provisional.

[Author's Response/Revision]

Thank you for your professional comments. We added a description of this result in the discussion section.

[Reviewer's Comment 8]

So if I understand correctly, this CNV involves the coding sequence and therefore subjects in possession of multiple copies produce an altered protein? and those with fewer copies a truncated protein?

[Author's Response/Revision]

We cannot confirm with certainty that CNV1 in possession of multiple copies produce an altered protein without in-depth studies of the mechanism. A CNV containing part of a gene's exon may affect the protein coding, transcription regulation, splicing regulation, RNA stability, and transport functions of the gene. The specific function depends on the sequence of the exon and its position within the gene regulatory network. This requires more in-depth mechanism study. (We have added an explanation of your suggestion in the discussion section)

[Reviewer's Comment 9]

Line 194 as reported before, you must indicate the statistical test applied.

Table 1 please confirm that loss means 1 single copy

Table 1 why A and not a?

[Author's Response/Revision]

Thank you for your comments. It has been corrected in the revision. And the LOSS has been defined in Section 2.5.

[Reviewer's Comment 10]

In addition to the points reported directly in the attached PDF, I invite you to discuss in depth the impact of CNV1 on the molecular structure of the gene involved, as this CNV seems to involve a good part of the coding sequence.

[Author's Response/Revision]

In the revision, we have added a discussion of the potential mechanisms of CNV1.

We would like to express our gratitude for your insightful comments, which have undoubtedly enhanced the quality and clarity of our manuscript. We believe that these revisions have significantly strengthened our work and addressed the concerns raised during the review process. Once again, we sincerely appreciate your time and effort in reviewing our manuscript. Your expertise and constructive feedback have been invaluable in shaping our research. We look forward to hearing from you regarding the final decision on our submission.

Round 2

Reviewer 3 Report

Comments and Suggestions for Authors

Dear Authors,

I have read version number 2 of the paper and I am partially satisfied with the changes you have made.

Compared to my notes you did not take into consideration:

1. the structure of the different isoforms of the plectin gene (line 112 of v2 and line 97 of v1)

2. a little discussion on the yak genome (line 169 v2 and line 147 v1)

3. in table 2 specify that loss means only 1 copy.

Finally, you did not take into consideration the fact that CNV1 is present in the coding sequence of PLECTIN and it is not clear how this CNV changes the structure of the protein (Figure 1). This is a very important aspect because if the coding sequence were altered, the presence of multiple copies would be associated with the destruction of the gene, regardless of the number of copies. 

Author Response

Dear Reviewer,

Thank you for taking the time to review our manuscript and for providing valuable feedback. We appreciate your thorough evaluation of our work and your suggestions for improvement. We have carefully considered your comments and have made the necessary revisions accordingly.

[Reviewer's Comment 1]

the structure of the different isoforms of the plectin gene (line 112 of v2 and line 97 of v1)

[Author's Response/Revision]

According to the sequence information of the PLEC gene in the NCBI database (https://www.ncbi.nlm.nih.gov/nuccore/NC_030821.1?report=genbank&from=81055388&to=81102834&strand=true), the goat PLEC gene does not undergo alternative splicing. In the response, we mentioned that ‘In order to display the similarities and differences of amino acid sequences more comprehensively, we did not select the isoform 1 of PLEC in each species according to the order of expression abundance, but selected the isoform with the highest length and gene coverage among different species for comparison.’This information has been added in revised manuscript V2.

 [Reviewer's Comment 2]

 a little discussion on the yak genome (line 169 v2 and line 147 v1)

[Author's Response/Revision]

Thank you for your comments. We have devoted a brief discussion (Line250 v2,). In the new revision, we have added more descriptions.

[Reviewer's Comment 3]

in table 2 specify that loss means only 1 copy.

[Author's Response/Revision]

In response to your comments, we have corrected the definition of lose type in all our tables.

[Reviewer's Comment 4]

Finally, you did not take into consideration the fact that CNV1 is present in the coding sequence of PLECTIN and it is not clear how this CNV changes the structure of the protein (Figure 1). This is a very important aspect because if the coding sequence were altered, the presence of multiple copies would be associated with the destruction of the gene, regardless of the number of copies. 

[Author's Response/Revision]

Thank you for your constructive comments. In the new revision, we added the analysis of the amino acids that may be encoded in the CNV-1 region, and used the SWISS-MODEL to predict the protein structure. We found that the amino acid sequence encoded by CNV-1 region might form a different protein structure compared with the complete goat plectin protein structure. However, current research methods are not sufficient to carry out experiments in goat PLEC protein. In the discussion, a detailed description is given, and the prediction results of SWISSS-MODEL are added as supplementary materials.

We would like to express our gratitude for your insightful comments, which have undoubtedly enhanced the quality and clarity of our manuscript. We believe that these revisions have significantly strengthened our work and addressed the concerns raised during the review process. Once again, we sincerely appreciate your time and effort in reviewing our manuscript. Your expertise and constructive feedback have been invaluable in shaping our research. We look forward to hearing from you regarding the final decision on our submission.
